# Creation of photocyclic vertebrate rhodopsin by single amino acid substitution

**Kazumi Sakai[1], Yoshinori Shichida[1,2], Yasushi Imamoto[1], Takahiro Yamashita[1]***

[1]Department of Biophysics, Graduate School of Science, Kyoto University, Kyoto, Japan; [2]Research Organization for Science and technology, Ritsumeikan University, Kusatsu, Japan

**Abstract** Opsins are universal photoreceptive proteins in animals and can be classified into three types based on their photoreaction properties. Upon light irradiation, vertebrate rhodopsin forms a metastable active state, which cannot revert back to the original dark state via either photoreaction or thermal reaction. By contrast, after photoreception, most opsins form a stable active state which can photoconvert back to the dark state. Moreover, we recently found a novel type of opsins whose activity is regulated by photocycling. However, the molecular mechanism underlying this diversification of opsins remains unknown. In this study, we showed that vertebrate rhodopsin acquired the photocyclic and photoreversible properties upon introduction of a single mutation at position 188. This revealed that the residue at position 188 contributes to the diversification of photoreaction properties of opsins by its regulation of the recovery from the active state to the original dark state.

## Editor's evaluation

This manuscript describes an investigation of the evolution of monostable rhodopsins, typically found in vertebrates. It highlights that single amino acid changes in vertebrate rhodopsins can create a partial bistable retinal pigment that can be photoconverted back to the ground state or it will slowly convert back to the ground state retinal isomer. The rationale for the experiments came from the discovery of a very interesting activation mechanism of the nonvisual pigment Opn5L1. This work has important implications for how our visual pigments have been optimized during evolution, and it contributes important insights into engineering bistable pigments for optogenetic applications.

*For correspondence:
yamashita.takahiro.4z@kyoto-u.ac.jp

**Competing interest:** The authors declare that no competing interests exist.

## Introduction

Opsins are photosensitive G-protein-coupled receptors and are universally found in diploblastic and triploblastic animals. All opsins share common structural elements including seven transmembrane domains and bind a light-absorbing chromophore, retinal, via a Schiff base linkage to Lys296 (based on the bovine rhodopsin numbering system) of opsin. Opsins function for both visual and nonvisual photoreception and are classified into several groups based on their amino acid sequence (*Shichida and Matsuyama, 2009*; *Koyanagi and Terakita, 2014*). Bovine rhodopsin is the best-studied opsin (*Yau and Hardie, 2009*) and it functions as a visual photoreceptive protein in the retina and binds 11-*cis* retinal in the dark. Photoisomerization of retinal to the all-*trans* form produces the meta II intermediate of rhodopsin, which couples with G protein. Meta II is a metastable active state and spontaneously converts to meta III (*Heck et al., 2003*). In addition, light irradiation of meta II induces the formation of meta III rather than the original dark state (*Bartl et al., 2001*; *Ritter et al., 2008*).

That is, the active state meta II very inefficiently converts back to the original dark state by photo-reaction or thermal reaction. These observations show that vertebrate rhodopsin is specialized for photoactivation, and is thus characterized as a monostable opsin. By contrast, mollusk and arthropod rhodopsins form a stable active state, the acid-meta state, by photoisomerization of 11-*cis* to all-*trans* retinal, and the active state can photoconvert back to the original dark state, which contains 11-*cis* retinal (*Koyanagi and Terakita, 2014*; *Yau and Hardie, 2009*). That is, these opsins have two stable states, the dark and active states, which are interconvertible by light and thus are known as bistable opsins. Recent accumulation of knowledge about the molecular properties of opsins revealed that many members of various opsin groups are bistable opsins (*Figure 1—figure supplement 1A*), which suggests that vertebrate rhodopsin evolved as a monostable opsin from an ancestral bistable opsin (*Shichida and Matsuyama, 2009*).

Recently, we identified a novel type of opsin, Opn5L1, as a photocycle opsin (*Sato et al., 2018*). Opn5L1 binds all-*trans* retinal, not 11-*cis* retinal, to form the active state in the dark. Light irradiation suppresses the G protein activation ability of Opn5L1 by the photoisomerization of the retinal to 11-*cis* form. Subsequent formation of a covalent adduct between the retinal and Cys188 of the opsin induces the conversion of the C11 = C12 double bond to a single bond in the retinal. Afterward, the thermal rotation of the C11–C12 single bond in the retinal results in dissociation of the Cys188-retinal adduct and regeneration of the original dark state. The combination of photoisomerization and thermal isomerization of retinal regulates the ability of Opn5L1 to activate G protein, making this the first animal opsin whose activity is controlled by its photocyclic reaction.

Comparison of the amino acid sequences among opsins shows that the cysteine residue at position 188 is well conserved in the Opn5L1 group but rarely found in other opsin groups (*Sato et al., 2018*; *Yamashita, 2020*), which supports the importance of Cys188 for the unique photocyclic reaction of Opn5L1. On the other hand, vertebrate rhodopsin and cone pigments, which are characterized as monostable opsins, have a glycine residue at this position (*Figure 1—figure supplement 1A*). In this study, we analyzed whether the mutation at position 188 (*Figure 1—figure supplement 1B*) can make bovine rhodopsin photocyclic. Our detailed analysis revealed that G188C mutant photoconverted to the active state, meta II, which thermally recovered to the original dark state. In addition, light irradiation of meta II of G188C mutant induced reversion to the original dark state. Therefore, G188C mutant of bovine rhodopsin exhibits the photocyclic and photoreversible property and the residue at position 188 regulates the recovery from the active state to the original dark state in vertebrate rhodopsin.

## Results and discussion
### Acquisition of photocyclic property of bovine rhodopsin G188C mutant

In a previous report, we revealed that Opn5L1 possesses a cysteine residue at position 188, which underlies the photocyclic reaction of the opsin (*Sato et al., 2018*). Thus, to analyze whether G188C mutant of bovine rhodopsin acquires the photocyclic property, we purified G188C mutant after reconstitution with 11-*cis* retinal. However, we found that G188C mutant has much lower thermal stability than wild-type. That is, G188C mutant gradually decayed during incubation in the dark at 37°C (*Figure 1B*), whereas wild-type was quite stable under the same conditions (*Figure 1A*). Therefore, we improved the thermal stability of G188C mutant to analyze the detailed molecular properties of the mutant. Following previous reports (*Xie et al., 2003*; *Standfuss et al., 2007*), we introduced two cysteine residues (N2C/D282C) into the mutant and measured the thermal decay rate during incubation in the dark at 37°C. The time-dependent spectral changes showed that this G188C/N2C/D282C mutant decayed much more slowly than G188C mutant (*Figure 1C*). Therefore, we compared the spectral changes among wild-type, N2C/D282C and G188C/N2C/D282C mutant at 20°C. The spectrum of wild-type was shifted into the UV region after yellow light irradiation (*Figure 1—figure supplement 2A*), which is indicative of the formation of meta II intermediate containing all-*trans*-15-*anti* retinal (*Figure 1D*). Subsequently, the absorbance at around 470 nm increased, which indicated the transition from meta II to meta III intermediate containing all-*trans*-15-*syn* retinal (*Vogel et al., 2003*; *Figure 1D*). These spectral changes were also observed in N2C/D282C (*Figure 1E*) as noted in the previous report (*Xie et al., 2003*). By contrast, G188C/N2C/D282C mutant had the absorption maximum ($\lambda_{max}$) at 487 nm and its spectrum was also shifted into the UV region to form meta II by

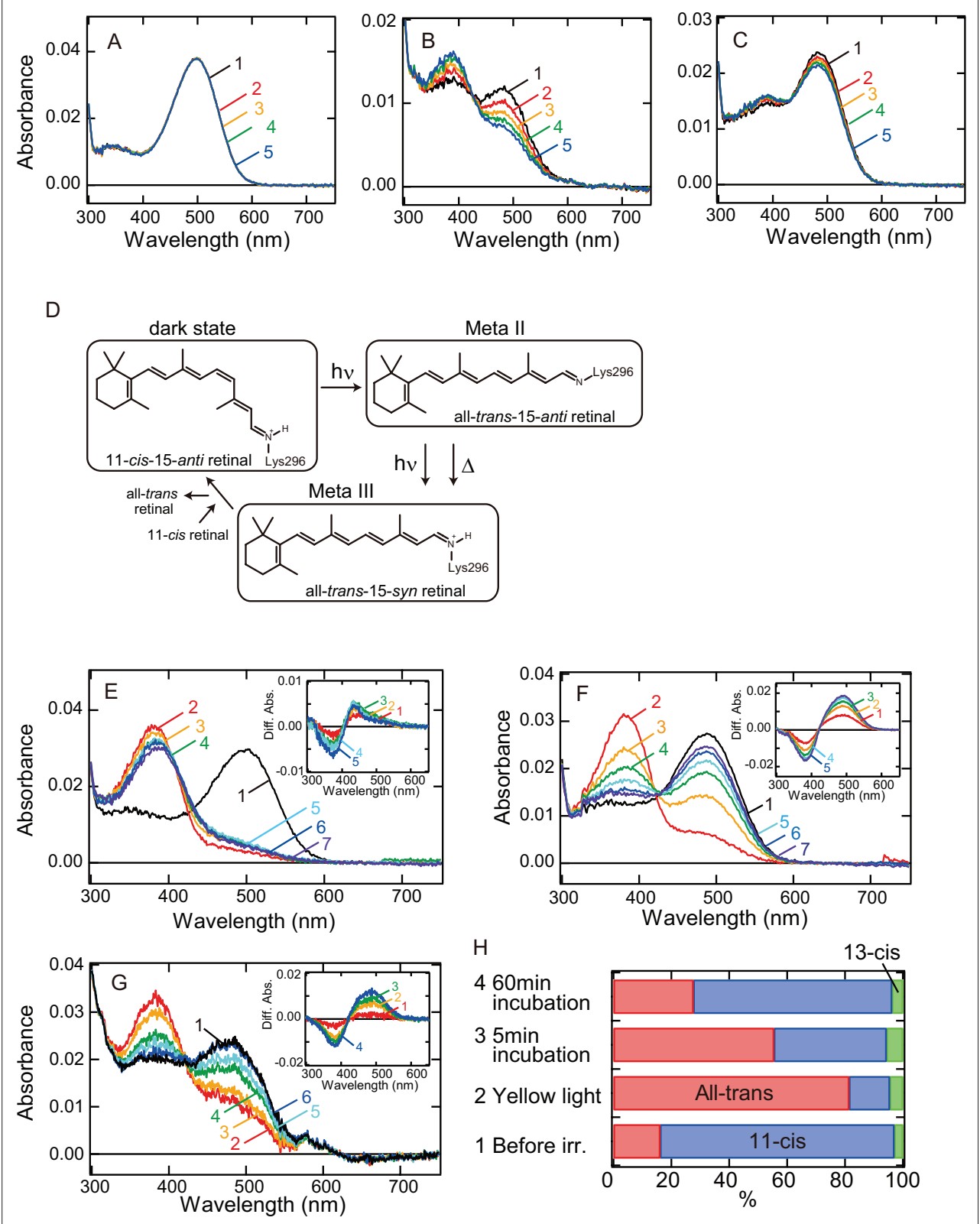

**Figure 1.** Thermal recovery of bovine rhodopsin G188C mutant after yellow light irradiation. Thermal stability of wild-type (**A**) and G188C (**B**) and G188C/N2C/D282C (**C**) mutants of bovine rhodopsin purified after the incubation with 11-*cis* retinal. Absorption spectra were recorded after 0, 5, 10, 15, and 20 min incubation (curves 1–5, respectively) in the dark at 37°C. (**D**) The schematic presentation of the retinal configuration change of wild-type bovine rhodopsin. The dark state, meta II, and meta III contain 11-*cis*-15-*anti* retinal, all-*trans*-15-*anti* retinal and all-*trans*-15-*syn* retinal, respectively

*Figure 1 continued on next page*

*Figure 1 continued*

(*Ritter et al., 2008*). Absorption spectra of N2C/D282C (**E**) and G188C/N2C/D282C (**F**) mutants of bovine rhodopsin purified after the incubation with 11-*cis* retinal. Spectra were recorded in the dark (curve 1) and 0, 5, 15, 30, 60, and 120 min after yellow light irradiation (curves 2–7, respectively) at 20°C. (Inset) Difference spectra obtained by subtracting the spectrum just after irradiation (curve 2 in (**E**) and (**F**)) from the spectra measured after irradiation (curves 3–7 in (**E**) and (**F**)) (curves 1–5, respectively). (**G**) Absorption spectra of G188C/N2C/D282C mutant measured at 37°C. Spectra were recorded in the dark (curve 1) and 0.1, 10, 50, 100, and 1000s after yellow flash light irradiation (curves 2–6, respectively). (Inset) Difference spectra obtained by subtracting the spectrum just after irradiation (curve 2 in (**G**)) from the spectra measured after irradiation (curves 3–6 in (**G**)) (curves 1–4, respectively). (**H**) Isomeric compositions of retinal of G188C/N2C/D282C mutant. The retinal configurations were analyzed by high-performance liquid chromatography (HPLC) after extraction of the chromophore from the samples before light irradiation and 0, 5, and 60 min after yellow light irradiation at 20°C as shown in *Figure 1—figure supplement 4*.

The online version of this article includes the following figure supplement(s) for figure 1:

**Figure supplement 1.** Characterization of the amino acid residue at position 188 of opsins.

**Figure supplement 2.** Thermal reaction of bovine rhodopsin wild-type and G188C mutant after yellow light irradiation.

**Figure supplement 3.** Thermal reaction of bovine rhodopsin Gly188 mutants after yellow light irradiation.

**Figure supplement 4.** High-performance liquid chromatography (HPLC) analysis of retinal configuration.

**Figure supplement 5.** Absorption spectra of Opn5L1 and bovine rhodopsin G188C mutant.

yellow light irradiation. During subsequent incubation in the dark, decreased absorbance in the UV region and a concomitant increase of the absorbance at around 485 nm were observed (*Figure 1F*). Analysis of the retinal configurations showed that light irradiation triggered the isomerization of the retinal to the all-*trans* form, which converted back to the 11-*cis* form during the subsequent incubation in the dark (*Figure 1H*). This interconversion of the retinal isomers can explain the spectral change of G188C/N2C/D282C mutant after light irradiation. Thermal recovery of the original dark state after light irradiation was also observed at 37°C (*Figure 1G*). In addition, G188C mutant showed thermal recovery of the absorption spectrum of the original dark state after light irradiation at 20°C (*Figure 1—figure supplement 2B*), which was confirmed by an increase of the amount of 11-*cis* retinal during the incubation after light irradiation (*Figure 1—figure supplement 2C*). Collectively, these results showed that G188C mutation led to acquisition of the ability to thermally recover the original dark state from the photoactivated state.

We also analyzed whether other G188 mutants acquire the photocyclic property. A previous study showed that G188E and G188R mutants of human rhodopsin cannot form the photopigments after reconstitution with 11-*cis* retinal (*Sung et al., 1993*). Thus, we introduced 16 other mutations at position 188 of bovine rhodopsin and prepared the mutant proteins purified after reconstitution with 11-*cis* retinal. We successfully detected the photopigments from eight of these mutants (curve 1 in *Figure 1—figure supplement 3*). $\lambda_{max}$ of the mutants was blue-shifted from that of wild-type (500 nm) with one exception, G188D (509 nm) (*Table 1*). Yellow light irradiation of these mutants shifted the spectra into the UV region to form meta II (curve 2 in *Figure 1—figure supplement 3*). During subsequent incubation in the dark at 20°C, each mutant showed characteristic spectral changes (curves 3–8 in *Figure 1—figure supplement 3*). However, these spectral changes were different from a substantial increase of the absorbance at around their $\lambda_{max}$. These results

**Table 1.** Comparison of $\lambda_{max}$ in the dark state and spectral components after UV light irradiation.

| | $\lambda_{max}$* | Dark state† (%) | Meta II† (%) | Meta III† (%) |
|---|---|---|---|---|
| Wild-type | 500 | 21.6‡ | 13.7 | 64.7 |
| G188C | 487 | 41.2‡ | 48.5 | 10.3 |
| G188A | 494 | 18.1 | 51.7 | 30.2 |
| G188D | 509 | 30.9 | 67.7 | 1.4 |
| G188M | 491 | 9.2 | 10.0 | 80.8 |
| G188N | 492 | 0 | 0 | 100 |
| G188Q | 493 | 2.4 | 5.5 | 92.1 |
| G188S | 495 | 14.6 | 37.7 | 47.7 |
| G188T | 488 | 11.0 | 2.9 | 86.1 |
| G188V | 486 | 7.3 | 63.4 | 29.3 |

*$\lambda_{max}$ was estimated from the absorption spectra shown in Figure 2 and Figure 2—figure supplement 1.

†Component ratios of the dark state, meta II, and meta III were calculated based on the spectral changes induced by UV light irradiation shown in Figure 2 and Figure 2—figure supplement 1.

‡Ratios of the dark state in wild-type and G188C are comparable to those of 11-*cis* retinal obtained by the retinal configuration analysis (24% in wild-type and 40% in G188C).

showed that the thermal recovery to the original dark state after light irradiation was not clearly detected in these mutants. Thus, we concluded that the photocyclic property was observed uniquely in G188C mutant.

## Acquisition of photoreversible property of bovine rhodopsin G188C mutant

We also analyzed whether meta II of G188C mutant converts back to the original dark state in a light-dependent manner. We cooled wild-type and G188C mutant to 0°C to prevent the thermal reaction of meta II and measured their spectral changes induced by yellow light and subsequent UV light irradiation. Yellow light irradiation of wild-type resulted in the formation of meta II, and subsequent UV light irradiation shifted the spectrum into the visible region with $\lambda_{max}$ (~470 nm) blue-shifted from that of the original dark state (*Figure 2A*). Previous studies revealed that this state is comparable to meta III (*Figure 1D*; *Bartl et al., 2001*; *Ritter et al., 2008*). We also constructed template absorption spectra of the dark state, meta II, and meta III modeled by the previous method (*Lamb, 1995*; *Govardovskii et al., 2000*; *Figure 2—figure supplement 1B*) and fitted the difference spectrum (curve 2 in the inset of *Figure 2A*) calculated by subtracting the spectrum after yellow light irradiation from that after UV light irradiation. Our fitting analysis showed that meta III was formed much more efficiently than the original dark state by UV light irradiation of meta II (*Figure 2—figure supplement 1C* and *Table 1*). This spectral analysis was consistent with the observation that UV light irradiation produced a very limited amount of 11-*cis* retinal from a large amount of all-*trans* retinal (*Figure 2C*). These results confirmed that UV light irradiation of meta II induces the syn/anti isomerization of the C = N double bond of the Schiff base more efficiently than it does the *cis/trans* isomerization of the retinal.

Yellow light irradiation of G188C mutant induced the formation of meta II, and subsequent UV light irradiation shifted the spectrum into the visible region with $\lambda_{max}$ quite similar to that of the original dark state (*Figure 2B*). Yellow light reirradiation caused formation of a state whose spectrum almost overlapped with that induced by the first yellow light irradiation (curve 4 in *Figure 2B*). Spectral changes induced by UV light irradiation and yellow light reirradiation were mirror images of each other (curves 2 and 3 in the inset of *Figure 2B*). We fitted the UV light-dependent spectral change with the template spectra and showed that the original dark state was formed much more efficiently than meta III by UV light irradiation of meta II (*Figure 2—figure supplement 1C* and *Table 1*). This was supported by the observation that UV light irradiation of G188C mutant increased the amount of 11-*cis* retinal more efficiently than UV light irradiation of wild-type (*Figure 2D*). These results suggested that meta II of G188C mutant can efficiently photoconvert back to the original dark state. Next, we measured the ability of G188C mutant to activate Gi-type of G protein, because bovine rhodopsin can activate not only transducin but also Gi/Go-types of G protein (*Yamashita et al., 2000*; *Terakita et al., 2002*). Our GTPγS-binding assay showed that the light-dependent Gi activation ability of G188C was equivalent to that of wild-type (*Figure 2E, F*). Subsequent UV light irradiation of G188C mutant suppressed the ability and yellow light reirradiation increased the ability (*Figure 2F*), which can be explained by the changes of the absorption spectra and the retinal configurations (*Figure 2B, D*). In addition, G188C/N2C/D282C mutant could also exhibit interconvertibility between the original dark state and meta II upon yellow light and UV light irradiation at 0°C (*Figure 2G*). These data showed that G188C mutant acquired the property of photoreversibility between the dark state and meta II. We also analyzed the photoreaction of eight other mutants. In all of these mutants, we observed the spectral shift to the UV region by yellow light irradiation and the reincrease of the absorbance in the visible region by subsequent UV light irradiation (*Figure 2—figure supplement 1A, C*). Fitting of the difference spectra calculated before and after UV light irradiation using template spectra of the dark state, meta II, and meta III (cyan curves in *Figure 2—figure supplement 1C*) provided information about the component ratios of the dark state, meta II, and meta III after UV light irradiation (*Table 1*). These results indicated that the recovery to the original dark state upon UV light irradiation occurs most efficiently in G188C mutant.

## Speeding up of the photocycle in G188C mutant

Next, we investigated whether the alteration of the lifetime of meta II can modulate the photocycle rate of G188C mutant. It has been reported that E122Q mutation of vertebrate rhodopsin accelerates the decay of meta II and thus shortens the lifetime of meta II (*Imai et al., 1997*; *Imai et al., 2007*).

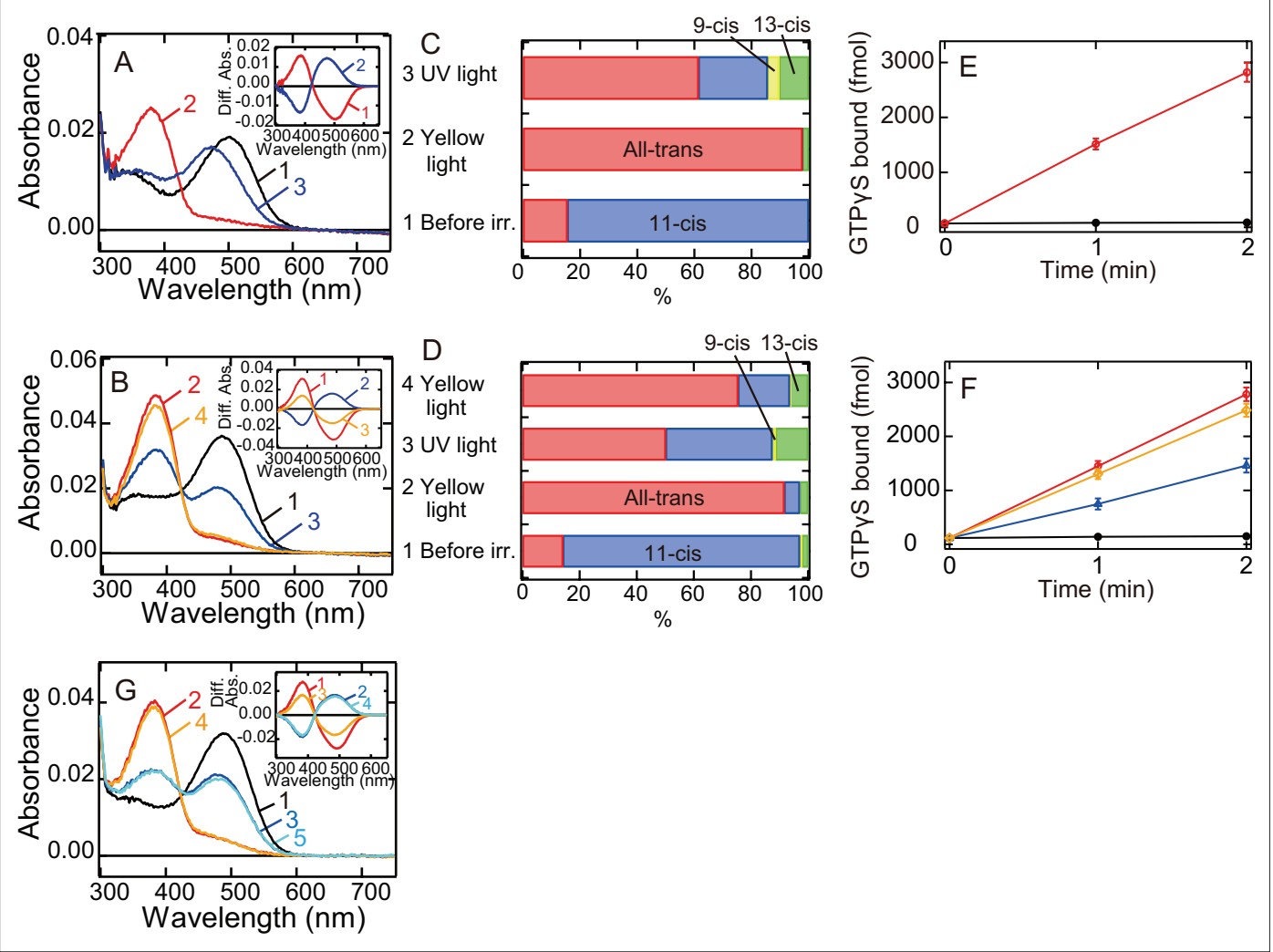

**Figure 2.** Photoreaction, retinal configuration, and G protein activation of bovine rhodopsin G188C mutant. Absorption spectra of wild-type (**A**) or G188C mutant (**B**) of bovine rhodopsin purified after the incubation with 11-*cis* retinal at 0°C. Spectra were recorded in the dark (curve 1), after yellow light (>500 nm) irradiation (curve 2), after subsequent UV light (360 nm) irradiation (curve 3) and after yellow light reirradiation (curve 4). (Inset) Spectral changes of wild-type (**A**) or G188C mutant (**B**) induced by yellow light irradiation (curve 1), subsequent UV light (curve 2) irradiation and yellow light reirradiation (curve 3). Difference spectra were calculated based on the spectra shown in (**A**) and (**B**). Isomeric compositions of retinal of wild-type (**C**) and G188C mutant (**D**). The retinal configurations were analyzed by high-performance liquid chromatography (HPLC) after extraction of the chromophore from the samples before light irradiation, after yellow light irradiation, after subsequent UV light irradiation and after yellow light reirradiation at 0°C as shown in *Figure 2—figure supplement 2*. (**E**) Gi-type of G protein activation ability of wild-type. The activation ability was measured in the dark (closed circle) and after yellow light irradiation (open circle). (**F**) Gi-type of G protein activation ability of G188C mutant. The activation ability was measured in the dark (closed circles), after yellow light irradiation (open circles), after subsequent UV light irradiation (open triangles) and after yellow light reirradiation (open diamonds). Data shown in (**E**) and (**F**) were obtained at 0°C and are presented as the means ± SEM of three independent experiments. (**G**) Absorption spectrum of G188C/N2C/D282C mutant purified after incubation with 11-*cis* retinal at 0°C. Spectra were recorded in the dark (curve 1), after yellow light (>500 nm) irradiation (curve 2), after subsequent UV light (360 nm) irradiation (curve 3), after yellow light reirradiation (curve 4) and after UV light reirradiation (curve 5). (Inset) Spectral changes induced by yellow light irradiation (curve 1), subsequent UV light (curve 2) irradiation, yellow light reirradiation (curve 3), and UV light reirradiation (curve 4). Difference spectra were calculated based on the spectra shown in (**G**).

The online version of this article includes the following figure supplement(s) for figure 2:

**Figure supplement 1.** Absorption spectra of bovine rhodopsin Gly188 mutants.

**Figure supplement 2.** High-performance liquid chromatography (HPLC) analysis of retinal configuration.

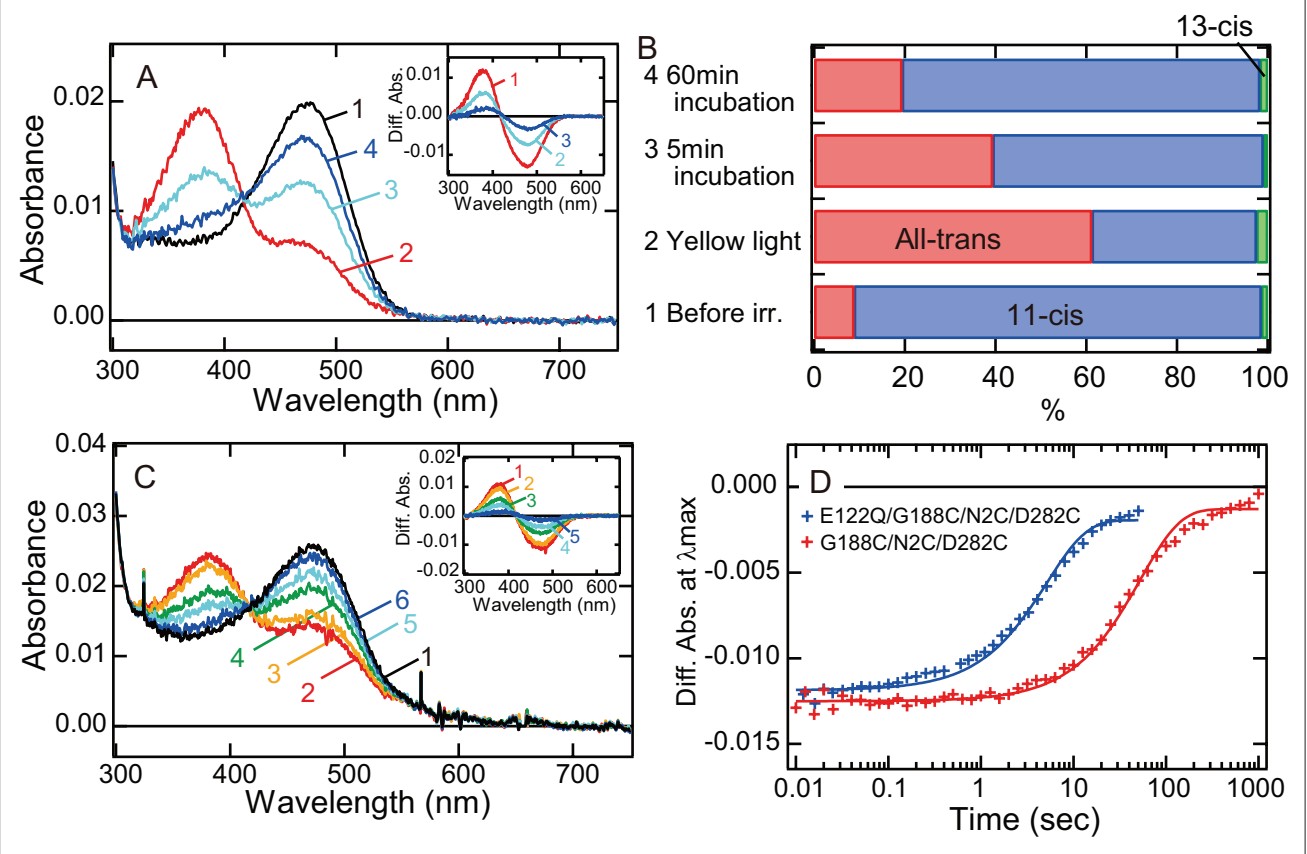

**Figure 3.** Faster recovery rate of the photocycle of bovine rhodopsin G188C mutant by introducing E122Q mutation. (**A**) Absorption spectra of E122Q/G188C/N2C/D282C mutant measured at 0°C. Spectra were recorded in the dark (curve 1) and 0, 5, and 60 min after yellow light (>500 nm) irradiation (curves 2–4, respectively). (Inset) Difference spectra obtained by subtracting the spectrum before irradiation (curve 1 in (**A**)) from the spectra measured after irradiation (curves 2–4 in (**A**)) (curves 1–3, respectively). (**B**) Isomeric compositions of retinal of E122Q/G188C/N2C/D282C mutant. The retinal configurations were analyzed by high-performance liquid chromatography (HPLC) after extraction of the chromophore from the samples before light irradiation and 0, 5, and 60 min after yellow light irradiation at 0°C as shown in **Figure 3—figure supplement 1**. (**C**) Absorption spectra of E122Q/G188C/N2C/D282C mutant measured at 37°C. Spectra were recorded in the dark (curve 1) and 0.1, 1, 5, 10, and 50 s after yellow flash light irradiation (curves 2–6, respectively). (Inset) Difference spectra obtained by subtracting the spectrum before irradiation (curve 1 in (**C**)) from the spectra measured after irradiation (curves 2–6 in (**C**)) (curves 1–5, respectively). (**D**) Comparison of the thermal recovery process between G188C/N2C/D282C (red) and E122Q/G188C/N2C/D282C (blue). Difference absorbance at $\lambda_{max}$ obtained by subtracting the spectrum before irradiation from the spectra measured after irradiation shown in **Figure 1G** and (**C**) was plotted against time elapsed after irradiation. The time constants of the thermal recovery to the dark state of G188C/N2C/D282C and E122Q/G188C/N2C/D282C mutants at 37°C were 57.4 and 5.1 s, respectively.

The online version of this article includes the following figure supplement(s) for figure 3:

**Figure supplement 1.** High-performance liquid chromatography (HPLC) analysis of retinal configuration.

Therefore, we prepared E122Q/G188C/N2C/D282C mutant and measured its spectral change after light irradiation. Our spectral and retinal configuration analyses confirmed the thermal recovery of the original dark state in E122Q/G188C/N2C/D282C mutant after light irradiation at 0°C (**Figure 3A, B**). In addition, the photocycle rate of E122Q/G188C/N2C/D282C mutant at 37°C (**Figure 3C**) was about 12 times faster than that of G188C/N2C/D282C mutant (**Figure 3D**). Thus, alteration of the lifetime of meta II by single mutation successfully speeded up the photocycle of G188C mutant.

## Modulation of G protein activation ability by the photocyclic property

We also investigated whether the acquisition of the photocyclic property by G188C mutation affects the G protein activation ability. As shown in **Figure 2**, light-dependent Gi activation ability was equivalent between wild-type and G188C mutant at 0°C, where substantial thermal recovery to the original dark state was not observed in G188C mutant. Thus, we measured the intracellular cAMP level in cultured cells using a cAMP biosensor (GloSensor) and compared the change of the luminescence

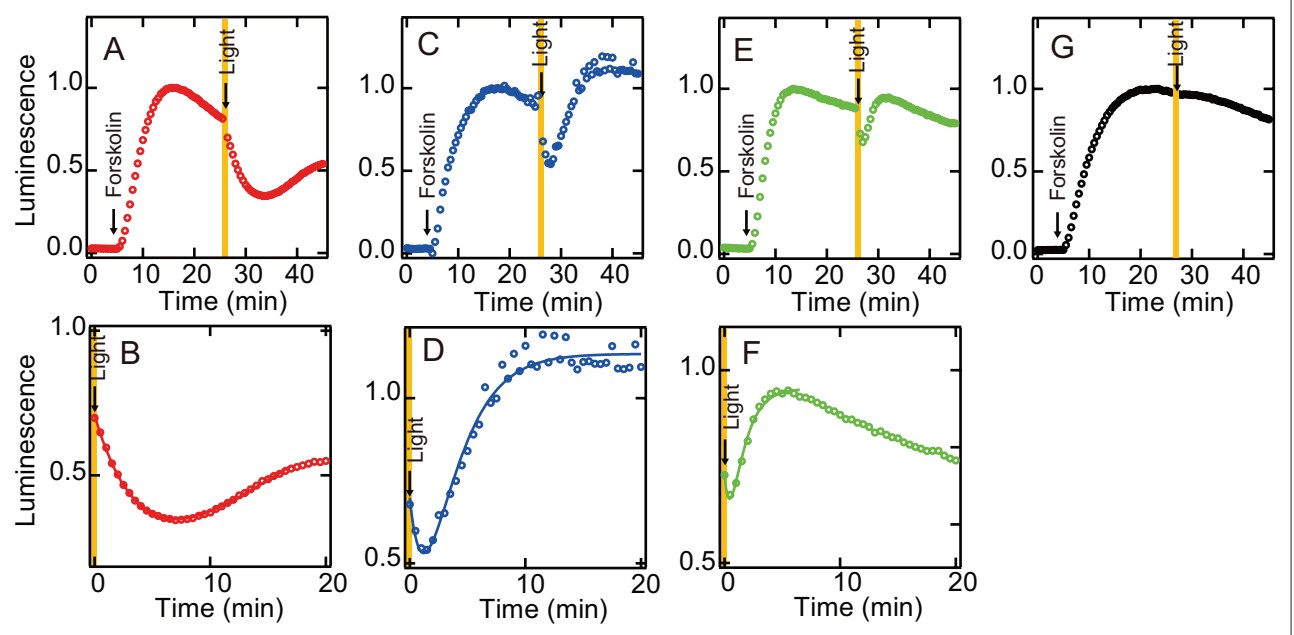

**Figure 4.** Light-mediated suppression of intracellular cAMP level by bovine rhodopsin mutants. The cAMP levels in N2C/D282C- (**A, B**), G188C/N2C/ D282C- (**C, D**), E122Q/G188C/N2C/D282C- (**E, F**), and mock- (**G**) transfected HEK293T cells were measured using the GloSensor cAMP assay at room temperature. The cells were incubated with 5 µM 11-*cis* retinal for 2 hr and subsequently treated with 2 µM forskolin prior to exposure to yellow light (>500 nm). Data were normalized to the maximum point before light irradiation. Detailed profiles of the light-dependent cAMP level changes in N2C/ D282C, G188C/N2C/D282C, and E122Q/G188C/N2C/D282C are shown in (**B**), (**D**), and (**F**), respectively.

from the biosensor triggered by bovine rhodopsin. The increase of the cAMP level induced by the addition of forskolin was attenuated by yellow light irradiation in the N2C/D282C bovine rhodopsin-transfected cells (*Figure 4A, B*), but not in the mock-transfected cells (*Figure 4G*), and it subsequently recovered slowly. By contrast, in the G188C/N2C/D282C mutant-transfected cells, we observed rapid recovery of the cAMP level after the decrease of the level induced by yellow light irradiation (*Figure 4C, D*). In addition, in the E122Q/G188C/N2C/D282C mutant-transfected cells, the cAMP level recovered more quickly from the decrease induced by yellow light irradiation (*Figure 4E, F*). These results showed that the acquisition of the photocyclic property by G188C mutation changes the G protein activation profile by promoting fast recovery to the original dark state.

## Formation of the photopigments by G188C mutant upon the addition of all-*trans* retinal

Finally, we analyzed whether G188C mutant forms the photopigments after reconstitution with all-*trans* retinal. We purified wild-type and G188C mutant after the addition of all-*trans* retinal to the suspension of rhodopsin-expressing cell membranes. The absorption spectrum of wild-type had almost no peaks in the visible and near-UV regions (*Figure 5A*). By contrast, the absorption spectrum of G188C mutant had a peak in the visible region (curve 1 in *Figure 5B*), which was derived from predominant incorporation of 11-*cis* and 9-*cis* retinals, not all-*trans* retinal (*Figure 5C*). Yellow light irradiation of this pigment resulted in conversion of the retinal to all-*trans* form to shift the spectrum into the UV region, and subsequent UV light irradiation reincreased the absorbance in the visible region by the isomerization of the retinal from the all-*trans* to the 11-*cis* form at 0°C (*Figure 5B, C*). This was quite similar to the finding for G188C mutant purified after reconstitution with 11-*cis* retinal (*Figure 2B, D*).

We also prepared the purified apo-proteins of N2C/D282C and G188C/N2C/D282C and investigated the process of regeneration of the photopigments upon the addition of 11-*cis* or all-*trans* retinal. The addition of 11-*cis* retinal to N2C/D282C and G188C/N2C/D282C quickly increased the absorbance at around 505 nm (*Figure 5D, E*) and 490 nm (*Figure 5H, I*), respectively, which showed the formation of their 11-*cis* retinal bound dark states. The addition of all-*trans* retinal to N2C/D282C resulted in a slight increase of the absorbance at around 480 nm (*Figure 5F, G*), whereas the addition

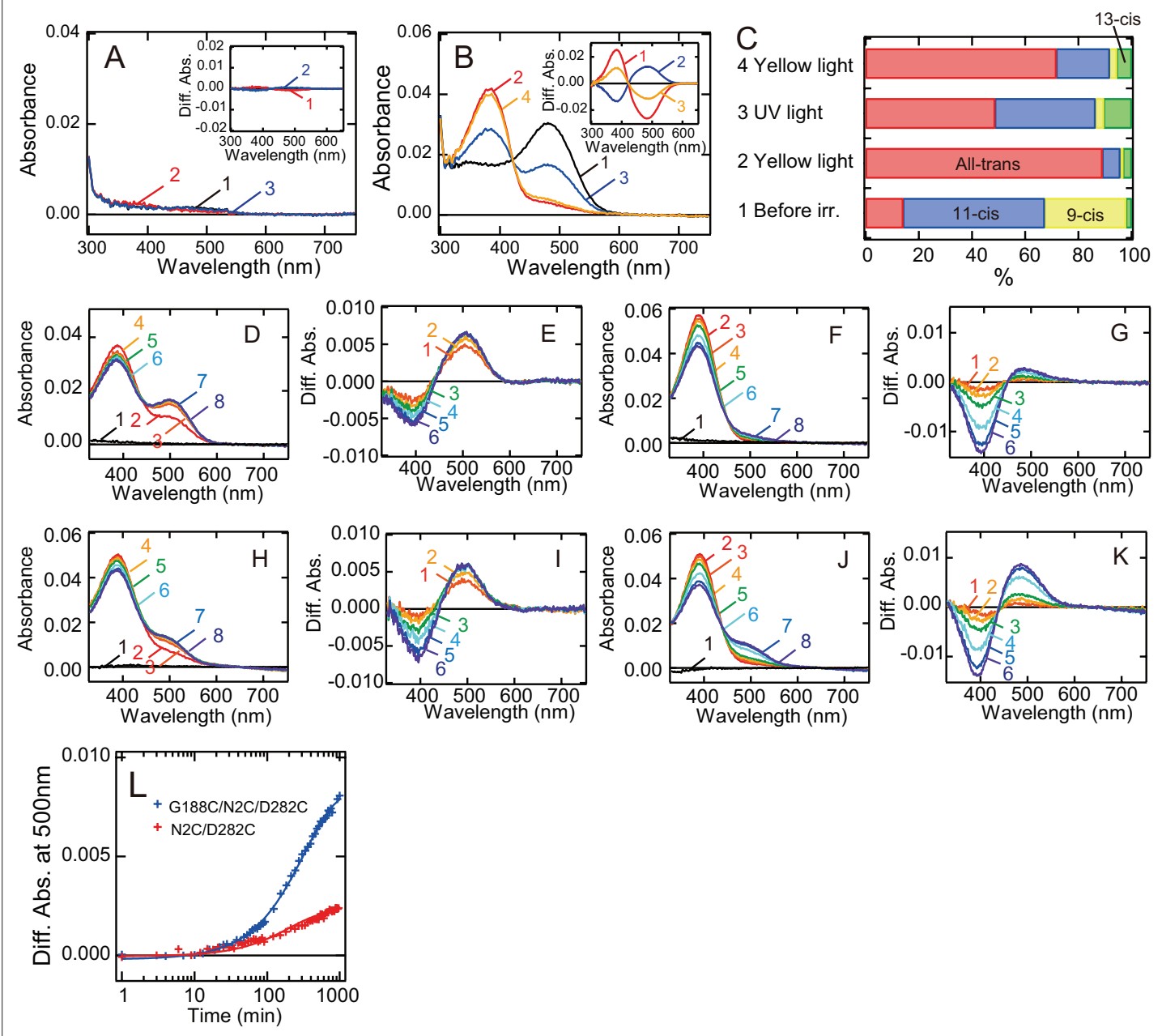

**Figure 5.** Formation of the photopigments of bovine rhodopsin G188C mutant after incubation with all-*trans* retinal. Absorption spectra of wild-type (**A**) or G188C mutant (**B**) purified after the addition of all-*trans* retinal to the suspension of rhodopsin-expressing cell membranes at 0°C. Spectra were measured in the dark (curve 1), after yellow light (>500 nm) irradiation (curve 2), after subsequent UV light (360 nm) irradiation (curve 3) and after yellow light reirradiation (curve 4). (Inset) Spectral change caused by yellow light irradiation (curve 1), subsequent UV light irradiation (curve 2) and yellow light reirradiation (curve 3). (**C**) Isomeric compositions of retinal of G188C mutant purified after the addition of all-*trans* retinal. The retinal configurations were analyzed by high-performance liquid chromatography (HPLC) after extraction of the chromophore from the samples before light irradiation, after yellow light irradiation, after subsequent UV light irradiation and after yellow light reirradiation shown in *Figure 5—figure supplement 1*. Regeneration of the photopigments by the addition of 11-*cis* (**D, E**) or all-*trans* (**F, G**) retinal to purified apo-protein of N2C/D282C. (**D**) Spectra were measured before (curve 1) and 0, 3, 6, 15, 30, 60, and 120 min after the addition of 1.1 µM 11-*cis* retinal (curves 2–8). (**E**) Difference spectra were calculated by subtracting the spectrum just after the addition of 11-*cis* retinal (curve 2 in (**D**)) from the spectra measured 3, 6, 15, 30, 60, and 120 min after the addition of 11-*cis* retinal (curves 3–8 in (**D**)) (curves 1–6, respectively). (**F**) Spectra were measured before (curve 1) and 0, 0.5, 1, 2, 6, 12, and 16 hr after the addition of 1.1 µM all-*trans* retinal (curves 2–8). (**G**) Difference spectra were calculated by subtracting the spectrum just after the addition of all-*trans* retinal (curve 2 in (**F**)) from the spectra measured 0.5, 1, 2, 6, 12, and 16 hr after the addition of all-*trans* retinal (curves 3–8 in (**F**)) (curves 1–6, respectively). Regeneration of the photopigments by the addition of 11-*cis* (**H, I**) or all-*trans* (**J, K**) retinal to purified apo-protein of G188C/N2C/D282C. (**H**) Spectra were measured before (curve 1) and 0, 3, 6, 15, 30, 60, and 120 min after the addition of 1.1 µM 11-*cis* retinal (curves 2–8). (**I**) Difference spectra were calculated by

*Figure 5 continued on next page*

Figure 5 continued

subtracting the spectrum just after the addition of 11-*cis* retinal (curve 2 in (**H**)) from the spectra measured 3, 6, 15, 30, 60, and 120 min after the addition of 11-*cis* retinal (curves 3–8 in (**H**)) (curves 1–6, respectively). (**J**) Spectra were measured before (curve 1) and 0, 0.5, 1, 2, 6, 12, and 16 hr after the addition of 1.1 µM all-*trans* retinal (curves 2–8). (**K**) Difference spectra were calculated by subtracting the spectrum just after the addition of all-*trans* retinal (curve 2 in (**J**)) from the spectra measured 0.5, 1, 2, 6, 12, and 16 hr after the addition of all-*trans* retinal (curves 3–8 in (**J**)) (curves 1–6, respectively). (**L**) Regeneration processes of the photopigments of N2C/D282C (red curve) and G188C/N2C/D282C (blue curve) by the addition of all-*trans* retinal as shown in (**F**) and (**J**) were monitored by the change of absorbance at 500 nm.

The online version of this article includes the following figure supplement(s) for figure 5:

**Figure supplement 1.** High-performance liquid chromatography (HPLC) analysis of retinal configuration.

of all-*trans* retinal to G188C/N2C/D282C resulted in a substantial increase of the absorbance at around 485 nm (*Figure 5J, K*). The regeneration ability of G188C/N2C/D282C in response to the addition of all-*trans* retinal was much higher than that of N2C/D282C (*Figure 5L*). These results showed that G188C mutant can uniquely form the photopigments upon the addition of not only 11-*cis* retinal but also all-*trans* retinal.

## Mechanism of conversion of the molecular property by G188C mutation

In general, vertebrate rhodopsin photoconverts to a metastable active state, meta II, by the *cis/trans* isomerization of the retinal and subsequently undergoes a thermal transition to meta III by the syn/anti isomerization of the C = N double bond of the Schiff base (*Figure 1D*; *Ritter et al., 2008*; *Vogel et al., 2003*; *Hofmann et al., 2009*). Thus, in response to light and heat, meta II exhibits greater conversion from all-*trans*-15-*anti* to all-*trans*-15-*syn* retinal than from all-*trans* to 11-*cis* retinal and recovers to the original dark state very inefficiently. By contrast, G188C mutant can revert to the original dark state from meta II much more efficiently than wild-type, possibly as a result of the preferential isomerization from all-*trans* to 11-*cis* retinal within the chromophore-binding pocket of meta II. The molecular models of the dark state and meta II of G188C mutant constructed based on the crystal structures of wild-type (*Choe et al., 2011*; *Okada et al., 2004*) suggest that Cys188 can be located in the vicinity of the C11 = C12 position of the retinal and Glu181 (*Figure 1—figure supplement 1B*). The protonation of the Schiff base in the dark state is neutralized by Glu113 (*Sakmar et al., 1989*; *Zhukovsky and Oprian, 1989*; *Nathans, 1990*), from which the counterion position switches to Glu181 in meta I (*Yan et al., 2003*; *Lüdeke et al., 2005*), a precursor of meta II. It should be noted that G188D mutant formed a substantial amount of meta I after light irradiation (*Figure 2—figure supplement 1A*), which can be explained by the stabilization of the protonated Schiff base of meta I by the introduction of an aspartic acid residue. Moreover, during the process of the formation of meta II, Cys188, Glu181, and the adjacent water molecule approach the Schiff base, whereas Glu113 and Ser186 move away from the Schiff base (*Figure 1—figure supplement 1B*). A previous study suggested that the syn/anti isomerization of the C = N double bond of the Schiff base in meta II can be promoted by the appropriate hydrogen-bonding network around the Schiff base as well as the specific steric chromophore–protein interaction (*Vogel et al., 2003*). Thus, we speculate that the cysteine residue introduced at position 188 disturbs the local structure and the hydrogen-bonding network around the Schiff base and Glu181 in meta II, which prevents the syn/anti isomerization of the C = N double bond of the Schiff base.

The addition of all-*trans* retinal to G188C mutant resulted in the formation of 9-*cis* or 11-*cis* retinal-containing photopigments (*Figure 5*). 9-*cis* or 11-*cis* retinal within G188C mutant would be formed from all-*trans* retinal on the outside or inside of the protein. If the thermal isomerization of the retinal occurs on the outside of the protein, we can expect the formation of the photopigments also from wild-type. However, this occurred much less efficiently in wild-type. Thus, we speculate that 9-*cis* or 11-*cis* retinal would be formed from all-*trans* retinal on the inside of G188C mutant, which is consistent with the acceleration of the thermal *cis/trans* isomerization of the retinal in meta II of G188C mutant.

Quite recently, we showed that T188C mutation of a bistable opsin, Opn5m, induces the thermal isomerization of the retinal from *cis* to *trans* isomer to make the opsin photocyclic (*Fujiyabu et al., 2022*). In this study, we observed the thermal recovery from the photoactivated state in G188C mutant (*Figure 1* and *Figure 1—figure supplement 2*), but not in other mutants (*Figure 1—figure supplement 3*). Thus, the cysteine residue at position 188 would have a special role to facilitate the thermal

isomerization of the retinal. In a previous study of Opn5L1, we revealed that the light-dependent adduct formation between Cys188 and 11-*cis* retinal accelerates the isomerization to all-*trans* retinal to recover the original dark state (*Sato et al., 2018*). During this process of photocycling of Opn5L1, our spectral analysis detected an increase of the absorbance at around 270 nm (*Figure 1—figure supplement 5A, B*), which is derived from the breaking of the retinal-conjugated double bond system by the adduct formation. However, in this study, the introduction of the cysteine residue at position 188 of bovine rhodopsin induced the isomerization from all-*trans* to 11-*cis* retinal, which is a reverse reaction to the thermal isomerization of the retinal in Opn5L1. Moreover, we could not observe a clear increase of the absorbance at around 270 nm during the thermal reaction of bovine rhodopsin G188C mutant after photoreception (*Figure 1—figure supplement 5C, D*). Thus, the detailed molecular mechanism of the acceleration of the thermal recovery to the original dark state in G188C mutant remains unknown, but one possible mechanism is that the cysteine residue introduced at position 188 transiently forms an adduct with all-*trans* retinal after photoactivation and quickly dissociates from the retinal after the isomerization to 11-*cis* retinal. An alternative possible mechanism can be predicted based on the analysis of channelrhodopsins. Channelrhodopsins have a unique cysteine residue near the C13 position of the retinal, which modulates the photocycle rate by regulating the syn/anti isomerization of the C = N double bond of the Schiff base without the adduct formation (*Ritter et al., 2013*; *Oppermann et al., 2019*). Considering this molecular mechanism in channelrhodopsins, the cysteine residue at position 188 near the C11 = C12 position of the retinal may possibly influence the structure and charge distribution of the retinal in meta II, which could trigger the thermal *cis*/*trans* isomerization of the retinal.

Comparison of the amino acid sequences reveals that most bistable opsins have a threonine or serine residue at position 188 (*Figure 1—figure supplement 1A*). Thus, we speculate that an ancestral bistable opsin possessed a threonine or serine residue at position 188, which was mutated into the glycine residue in the common ancestor of monostable opsins, including vertebrate rhodopsin, cone visual pigments, and pinopsin. Also, the mutation at position 188 may have contributed to the change from the bistable property to the monostable property in the evolutionary ancestor of vertebrate rhodopsin. This is consistent with our recent finding that mutations at Thr188 of a bistable opsin, Opn5m, drastically hamper the bistable photoreaction (*Fujiyabu et al., 2022*). However, we could not create a bistable opsin by a single mutation, G188T or G188S, of bovine rhodopsin (*Figure 2— figure supplement 1* and *Table 1*), which means that additional amino acid residue(s) are necessary to explain the difference between the bistable and monostable property from the viewpoint of the molecular evolution of opsins.

## Conclusion

In this study, we analyzed a series of mutants at position 188 of bovine rhodopsin and found that G188C mutant has a unique active state which can revert to the original dark state both by a thermal reaction and by a photoreaction. These results showed that the molecular properties of vertebrate rhodopsin can be converted to photocyclic and photoreversible properties by this single mutation. Little attention has been paid to the functional role of the residue at position 188 in opsins so far. The combination of the mutation at position 188 with other mutations in various opsins could perturb the local structure around the Schiff base, which could lead to interconversion of the molecular properties among opsins. Moreover, G188C mutant of bovine rhodopsin has several advantages as an optogenetic tool, because G188C mutant can be reconstituted in the presence of all-*trans* retinal and exhibits the photocyclic property, like channelrhodopsin (*Deisseroth and Hegemann, 2017*) in addition to its high expression yield in mammalian cultured cells and high G protein activation ability. We successfully showed that the change of the lifetime of meta II by the single mutation can modulate the photocycle rate of G188C mutant. Further accumulation of evidence about mutants of vertebrate rhodopsin can help to guide the modification of the molecular properties of G188C mutant, which will provide a novel type of optogenetic tools based on vertebrate rhodopsin.

## Materials and methods

**Key resources table**

| Reagent type (species) or resource | Designation | Source or reference | Identifiers | Additional information |
|---|---|---|---|---|
| Commercial assay or kit | In-Fusion HD Cloning | Clontech | Clontech: 639,647 | |
| Commercial assay or kit | GloSensor cAMP Assay Kit | Promega | Promega: E1290 | |
| Chemical compound, drug | Forskolin | FUJIFILM (Wako) | FUJIFILM:067-02191 | |
| Chemical compound, drug | n-Dodecyl-β-D-maltoside | DOJINDO | DOJINDO: D316 | |
| Chemical compound, drug | [$^{35}$S]GTPγS | PerkinElmer | PerkinElmer: NEG030H | |
| Software, algorithm | Igor Pro Ver. 6 | https://www.wavemetrics.com/ | | |
| Software, algorithm | PyMOL Ver. 1.1 | https://pymol.org/2/ | | |

## Preparation of bovine rhodopsin mutants

The mutant cDNAs of bovine rhodopsin (accession no. AB062417) were constructed using an In-Fusion cloning kit (Clontech). The wild-type and mutant cDNAs of bovine rhodopsin were inserted into the mammalian expression vector pUSRα (*Kayada et al., 1995*) or pCAGGS (*Niwa et al., 1991*). HEK293T cells were kindly provided by Dr. Satoshi Koike (Tokyo Metropolitan Institute of Medical Science, Tokyo Japan) (*Onishi et al., 1999*) and were authenticated by short tandem repeat profiling. The cells tested negative for mycoplasma contamination. The plasmid was transfected into HEK293T cells using the calcium-phosphate method. After 2-day incubation, the transfected cells were collected by centrifugation and suspended in Buffer A (50 mM HEPES, 140 mM NaCl, 3 mM MgCl$_2$, pH 6.5), and 11-*cis* or all-*trans* retinal was added to the cell suspension to reconstitute the photopigments. They were solubilized in Buffer A containing 1% dodecyl maltoside (DDM) and adsorbed to a Rho1D4 (anti-bovine rhodopsin monoclonal antibody) affinity column to purify the pigments. After washing the column with Buffer A containing 0.02% DDM, the pigment was eluted by adding synthetic peptide with the epitope sequence. To purify the apo-proteins of rhodopsin, the transfected cell membranes without the addition of retinal were solubilized in Buffer A containing 1% DDM and adsorbed to a Rho1D4 affinity column.

## Spectroscopic measurements

UV/Vis absorption spectra were recorded with a UV–visible spectrophotometer (UV-2450 and UV-2400, Shimadzu). Samples were kept at 0, 20, or 37°C using a cell holder equipped with a temperature-controlled circulating water bath in order to analyze the thermal reaction of the pigments in detail. The samples were irradiated with either yellow light through a Y-52 cutoff filter (Toshiba) or UV light through a UVD-36 glass filter (AGC Techno Glass) from a 1 kW tungsten halogen lamp (Master HILUX-HR; Rikagaku).

To monitor the process of the photocycle of G188C mutant of bovine rhodopsin, a time-resolved CCD spectrophotometer (C10000 system, Hamamatsu Photonics) was used (*Sakai et al., 2012*). Spectra were taken from G188C mutant samples in the dark and at different time points after irradiation (170-μs, yellow light through a Y-52 cutoff filter from a Xenon flash lamp). The temperature of the sample was kept at 37°C by a temperature controller (pqod, QUANTUM Northwest). Absorbance changes at $\lambda_{max}$ were plotted as a function of time and fitted with a single-exponential function to obtain the time constants for the recovery to the original dark state.

## Retinal configuration analysis

Retinal configurations within rhodopsin samples were analyzed by high-performance liquid chromatography (LC-10ATvp; Shimadzu) with a silica column (YMC-Pack SIL, particle size 3 μm, 150 × 6.0 mm, YMC) as previously described (*Tsutsui et al., 2007*).

## G protein activation assay

The activation of Gi-type of G protein was measured by GDP/GTPγS exchange of G protein using a radionucleotide filter-binding assay (*Yamashita et al., 2000*; *Yamashita et al., 2010*). Giαβγ was prepared by mixing rat Giα1 expressed in *Escherichia coli* strain BL21 (*Lee et al., 1994*) with Gtβγ purified from bovine retina (*Tachibanaki et al., 1997*). All of the assay procedures were carried out at 0°C. The assay mixture consisted of 10 nM pigment, 600 nM G protein, 50 mM HEPES (pH 7.0), 140 mM NaCl, 5 mM $MgCl_2$, 1 mM DTT, 0.01% DDM, 1 µM [$^{35}$S]GTPγS, and 2 µM GDP. Bovine rhodopsin wild-type and G188C mutant purified after reconstitution with 11-*cis* retinal were mixed with G protein solution and were kept in the dark or irradiated with yellow light (>500 nm) for 1 min, with subsequent UV light for 1 min or with yellow light reirradiation for 1 min. After irradiation, the GDP/GTPγS exchange reaction was initiated by the addition of [$^{35}$S]GTPγS solution to the mixture of rhodopsin and G protein. After incubation for the selected time in the dark, an aliquot (20 µl) was removed from the sample into 200 µl of stop solution (20 mM Tris/Cl [pH 7.4], 100 mM NaCl, 25 mM $MgCl_2$, 1 µM GTPγS, and 2 µM GDP), and it was immediately filtered through a nitrocellulose membrane to trap [$^{35}$S]GTPγS bound to G proteins. The amount of bound [$^{35}$S]GTPγS was quantitated by assaying the membrane with a liquid scintillation counter (Tri-Carb 2910 TR; PerkinElmer).

## cAMP level measurement in cultured cells

cAMP levels in HEK293T cells were measured using the GloSensor cAMP assay (Promega) according to the manufacturer's instructions and a previous report (*Bailes and Lucas, 2013*). HEK293T cells were seeded in 96-well plates at a density of 20,000 cells/well in low serum medium (D-MEM/F12 containing 0.25% FBS). After 24 hr of incubation, cells were transfected with 50 ng of rhodopsin plasmid and 50 ng of Glosensor 22F plasmid per well by the polyethylenimine transfection method. After overnight incubation, the medium was replaced with an equilibration medium which contained a 2% dilution of the GloSensor cAMP reagent stock solution, 10% FBS and 5 µM retinal in a $CO_2$-independent medium (Thermo Fisher Scientific). Following 2-hr equilibration at room temperature, luminescence from the cells was measured using a microplate reader (SpectraMax L, Molecular Devices). For the measurement of Gi activation by wild-type and mutant rhodopsin, the cells were first treated with 2 µM forskolin to increase the cAMP-dependent luminescence to the plateau level and subsequently stimulated for 30 s with yellow light through a Y-52 cutoff filter from a 1 kW tungsten halogen lamp.

## Acknowledgements

We thank Dr. Elizabeth Nakajima for a critical reading of the manuscript. We also thank Prof. Robert S Molday for the generous gift of a Rho1D4-producing hybridoma and Dr. Keita Sato for valuable discussion of the manuscript and technical advice about the cAMP assay. Funding: This work was supported in part by Grants-in Aid for Scientific Research of MEXT to YS (16H02515), YI (19K21848), and TY (16K07437), CREST, JST JPMJCR1753 (TY), a grant from the Kyoto University Foundation (TY) and a grant from the Takeda Science Foundation (TY).

## Additional information

### Funding

| Funder | Grant reference number | Author |
|---|---|---|
| Ministry of Education, Culture, Sports, Science and Technology | 16H02515 | Yoshinori Shichida |
| Ministry of Education, Culture, Sports, Science and Technology | 19K21848 | Yasushi Imamoto |
| Ministry of Education, Culture, Sports, Science and Technology | 16K07437 | Takahiro Yamashita |

| Funder | Grant reference number | Author |
|---|---|---|
| Japan Science and Technology Agency | JPMJCR1753 | Takahiro Yamashita |
| Kyoto University Education and Research Foundation | | Takahiro Yamashita |
| Takeda Science Foundation | | Takahiro Yamashita |

The funders had no role in study design, data collection, and interpretation, or the decision to submit the work for publication.

## Author contributions

Kazumi Sakai, Conceptualization, Data curation, Formal analysis, Investigation, Writing – original draft, Writing – review and editing; Yoshinori Shichida, Conceptualization, Formal analysis, Funding acquisition, Writing – original draft, Writing – review and editing; Yasushi Imamoto, Funding acquisition, Methodology, Writing – review and editing; Takahiro Yamashita, Conceptualization, Data curation, Formal analysis, Funding acquisition, Investigation, Project administration, Supervision, Writing – original draft, Writing – review and editing

## Author ORCIDs

Kazumi Sakai ⬤ http://orcid.org/0000-0001-6631-8546
Yasushi Imamoto ⬤ http://orcid.org/0000-0002-0803-4163
Takahiro Yamashita ⬤ http://orcid.org/0000-0002-7956-9288

## Decision letter and Author response

Decision letter https://doi.org/10.7554/eLife.75979.sa1
Author response https://doi.org/10.7554/eLife.75979.sa2

# Additional files

## Supplementary files

• Transparent reporting form

## Data availability

All data needed to evaluate the conclusions are present in the paper. The datasets of the current study are available in the Dryad repository (https://doi.org/10.5061/dryad.c866t1g88).

The following dataset was generated:

| Author(s) | Year | Dataset title | Dataset URL | Database and Identifier |
|---|---|---|---|---|
| Yamashita T | 2022 | Data from: Creation of photocyclic vertebrate rhodopsin by single amino acid substitution | http://dx.doi.org/10.5061/dryad.c866t1g88 | Dryad Digital Repository, 10.5061/dryad.c866t1g88 |

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
