## [Editor Report]

This manuscript describes an investigation of the evolution of monostable rhodopsins, typically found in vertebrates. It highlights that single amino acid changes in vertebrate rhodopsins can create a partial bistable retinal pigment that can be photoconverted back to the ground state or it will slowly convert back to the ground state retinal isomer. The rationale for the experiments came from the discovery of a very interesting activation mechanism of the nonvisual pigment Opn5L1. This work has important implications for how our visual pigments have been optimized during evolution, and it contributes important insights into engineering bistable pigments for optogenetic applications.

---

## [Decision Letter]

**Decision letter after peer review:**

Thank you for submitting your article "Creation of photocyclic vertebrate rhodopsin by single amino acid substitution" for consideration by *eLife*. Your article has been reviewed by 3 peer reviewers, and the evaluation has been overseen by a Reviewing Editor and Richard Aldrich as the Senior Editor. The following individuals involved in review of your submission have agreed to reveal their identity: Peter Hegemann (Reviewer #1); Gebhard Schertler (Reviewer #2).

Essential revisions:

Please see the discussion below regarding clarity and presentation of data. While no new experiments are necessary, the reviewers felt that improvements to the text would significantly enhance readability and breadth of appeal.

*Reviewer #1 (Recommendations for the authors):*

The manuscript of Sakai et al., reports about the long-standing question what internal constrains discriminate a cyclic rhodopsin kinetics from non-cyclic kinetics. This question has been discussed over many decades of rhodopsin research.

Based on the authors report about chicken OPN5L1 (Ref.7), which forms a thioadduct between a reactive Cys and the retinal upon illumination, they introduced such a Cys into the homologous position of bovine rhodopsin, G188C, rendering the naturally non-cyclic photoreaction cyclic. The photoproduct Meta II does not release the at-retinal and the rhodopsin converts back to the dark state. Other residues at this position were only partially active or unstable. At low temperature Meta II is further stabilized but may be photoconverted back to the dark state. This sounds trivial but it is not because wt rhodopsin Meta II state photoconverts it to Meta III with 15=N syn geometry thermally or photochemically and does not recover to the dark state.

The authors further stabilized the G188C-photoproducts by introducing two more Cys as reported already by Dan Oprian et al., in 2003. The triple mutant showed a further stabilized Meta II photoproduct whereas additional E122Q mutation reaccelerated it.

Finally, the authors showed that G188C mutants can also be reconstituted with at-retinal which is a great advantage over wt for potential optogenetic applications.

The G188C variants are fully active in terms of G-protein binding and the cAMP assay revealed the thermal photocycle kinetics of wt and mutants (Figure 4).

Comments:

What the reader also need to show are the differences between dark state and Meta II which could be visualized easily by an overlay of the two available structures.

The main question for the reader is to what extent can the principle of cyclization by introducing a Cys homologous positions be extended to other vertebrate rhodopsins. If this would be the case it would be a spectacular observation and open a new field for application. Now, it still looks exotic and as a exceptional observation.

The spectra of G188C at 37degC are of poor quality compared to the double or triple mutants in Figure 1 or the G188C spectrum in Figure S2. I would show the 20degC spectra in Figure 1.

The non-specialized reader would appreciate a scheme of cis-trans and "syn/anti" isomerization and if I interpret Figure S7 correctly is should be an anti/syn illumination because dark state and Meta II should be anti? Please clarify.

Figure S7 is helpful but should be introduced earlier in the text to make clear where C188 is located. I really do expect a deeper discussion of this issues including other relevant residues in Figure S7.

Interpretation

The weaker part of the manuscript is the interpretation of why G188C mutant prevents retinal release during the Meta II state and its thermal or photochemical anti/syn isomerization. Many invertebrate rhodopsins do not release the retinal during the Meta II state and do not have any Cys residue at this location. Moreover, some of them deprotonate during the meta state and others do not. The authors are experts in invertebrate rhodopsins as well and privileged to give a better interpretations.

Next, the suggestion that Cys causes transient thioadduct formation during the photocycle is not justified by anything and should be proposed with more care. For example a Cys at a similar position has been found in microbial Channelrhodopsins (ChRs). Substitution of this Cys slows down the photocycle dramatically and prevents syn/anti isomerization as it occurs during light adaptation. On the other hand some rhodopsin do not show such an anti/syn isomerization at all but removal of the Cys near the C13 position promotes anti/syn isomerization as found for example in Mermaid ChRs (Oppermann et al.,). The sulfur of the Cys seems to act as a nucleophile for retinal polyene chain, forming a thioadduct as in OPN5L or not as in Channelrhodopsin. But the Cys obviously influences the charge distribution in darkness or during the excited state. Moreover, residues in the active site including E113, E188 and other residues could act as steric constraints that influence the isomerization specificity as well.

I really do expect deeper discussion of this issues including other relevant residues in Figure S7.

What the reader also needs to know are the differences between dark state and Meta II which could be visualized by an overlay of the two available structures.

*Reviewer #2 (Recommendations for the authors):*

I support the publication without further modifications.

*Reviewer #3 (Recommendations for the authors):*

The data are thorough and convincing. However, the quality of presentation must be significantly improved.

1. Overall the manuscript is too long, with numerous unnecessary repetitions. It can be shortened by 30-40% without losing the message.

2. Evolutionary analysis (Figure S1) suggests that Ala, Ser, or Thr are in 188 position of most bistable opsins. How do the authors interpret this in the context of their data? The authors should discuss the difference in the properties of vertebrate opsin with Ala, Ser, or Thr are in 188 position (Figure S4), as compared to 188Cys? These data suggest that the residue in position 188 is not the only determinant of rhodopsin properties. The authors might want to revise Figure S7 accordingly.

3. Extensive editing, preferably by a native speaker, is needed. Some examples: line 35, "photoreceptions" should be "photoreception"; line 36, "sequences" should be "sequence"; line 50, "reversible with each other by light irradiations" should be "interconvertible by light"; lines 73, 86, 128, 148, 209, 223, delete "or not"; lines 73-74, "change the molecular property of bovine rhodopsin to acquire the photocyclic property" should be "can make bovine rhodopsin photocyclic"; line 91, "condition" should be "conditions"; line 125, "Altogether" should be "Collectively"; lines 152, 195 "irradiations" should be "irradiation"; lines 174-175, "overlapped that" should be "overlapped with that"; etc.

---

## [Author Response]

Reviewer #1 (Recommendations for the authors):[…]Comments:What the reader also need to show are the differences between dark state and Meta II which could be visualized easily by an overlay of the two available structures.

According to your comment, we constructed the structural models of the dark state and meta II of G188C mutant based on the 3D structures of the dark state and meta II of wild-type and superimposed these models to visualize the structural difference around position 188. We showed these structural models in Figure 1—figure supplement 1B.

The main question for the reader is to what extent can the principle of cyclization by introducing a Cys homologous positions be extended to other vertebrate rhodopsins. If this would be the case it would be a spectacular observation and open a new field for application. Now, it still looks exotic and as a exceptional observation.

I thank you very much for this important comment. Quite recently, we showed that a vertebrate non-visual bistable opsin, Opn5m, acquired the photocyclic property by the introduction of a cysteine residue at position 188 (Fujiyabu et al., (2022) Commun. Biol. 5, 63). Thus, we think that several opsins, including vertebrate rhodopsin and Opn5m, have the potential to convert into the photocyclic opsin by the introduction of a cysteine residue at position 188. Now we are systematically analyzing which opsins can acquire the photocyclic property by the introduction of a cysteine residue at position 188. To show our recent results, we added the following sentence in the “Results and discussion” section.

Page 13: “Quite recently, we showed that T188C mutation of a bistable opsin, Opn5m, induces the thermal isomerization of the retinal from cis to trans isomer to make the opsin photocyclic.”

The spectra of G188C at 37degC are of poor quality compared to the double or triple mutants in Figure 1 or the G188C spectrum in Figure S2. I would show the 20degC spectra in Figure 1.

The poor quality of the spectra of triple mutant (G188C/N2C/D282C) recorded at 37degC (Figure 1G) is due to the spectral measurement using a CCD spectrophotometer, in which we weakened the intensity of the monitoring light to prevent the rhodopsin bleaching in the sample. We already showed the spectra of the triple mutant at 20degC in Figure 1F, which were recorded by a scanning spectrophotometer with a single monochromator. We would like to show the spectra at 37degC in Figure 1G to confirm that the photocyclic reaction of G188C mutant can be observed at 37degC.

The non-specialized reader would appreciate a scheme of cis-trans and "syn/anti" isomerization and if I interpret Figure S7 correctly is should be an anti/syn illumination because dark state and Meta II should be anti? Please clarify.

According to your comment, we added a new figure (Figure 1D) to clarify the retinal configuration change of wild-type bovine rhodopsin.

Figure S7 is helpful but should be introduced earlier in the text to make clear where C188 is located. I really do expect a deeper discussion of this issues including other relevant residues in Figure S7.InterpretationThe weaker part of the manuscript is the interpretation of why G188C mutant prevents retinal release during the Meta II state and its thermal or photochemical anti/syn isomerization. Many invertebrate rhodopsins do not release the retinal during the Meta II state and do not have any Cys residue at this location. Moreover, some of them deprotonate during the meta state and others do not. The authors are experts in invertebrate rhodopsins as well and privileged to give a better interpretations.Next, the suggestion that Cys causes transient thioadduct formation during the photocycle is not justified by anything and should be proposed with more care. For example a Cys at a similar position has been found in microbial Channelrhodopsins (ChRs). Substitution of this Cys slows down the photocycle dramatically and prevents syn/anti isomerization as it occurs during light adaptation. On the other hand some rhodopsin do not show such an anti/syn isomerization at all but removal of the Cys near the C13 position promotes anti/syn isomerization as found for example in Mermaid ChRs (Oppermann et al.,). The sulfur of the Cys seems to act as a nucleophile for retinal polyene chain, forming a thioadduct as in OPN5L or not as in Channelrhodopsin. But the Cys obviously influences the charge distribution in darkness or during the excited state. Moreover, residues in the active site including E113, E188 and other residues could act as steric constraints that influence the isomerization specificity as well.I really do expect deeper discussion of this issues including other relevant residues in Figure S7.

I thank you very much for the valuable comment about the molecular mechanism underlying the photocyclic reaction in G188C mutant. The crystal structures of the dark state and meta II of bovine rhodopsin suggest that the appropriate hydrogen-bonding network and the specific steric interactions around the Schiff base regulate the deprotonation/reprotonation of the Schiff base and the syn/anti isomerization of the C=N double bond of the Schiff base during the formation and the decay of meta II. However, in the present situation, it is very difficult to clearly construct a molecular model to explain the acceleration of the syn/anti isomerization of the C=N double bond in meta II. In this study, we can speculate that the introduction of a cysteine residue at position 188 would disturb the local environment around the Schiff base and Glu181 in meta II, which prevents the syn/anti isomerization of the C=N double bond. However, in the previous version of the manuscript, a lack of detailed structural information about G188C mutant caused us to hesitate to speculate about the molecular mechanism underlying the photocyclic reaction in G188C mutant except for the acquisition of the special structural mechanism, the adduct formation. During the revision process, you kindly provided valuable information about channelrhodopsins indicating that a unique cysteine residue is located near the C13 position of the retinal and modulates the photocycle rate by regulating the syn/anti isomerization of the C=N double bond of the Schiff base. Thus, based on the comparison of the predicted structures of the dark state and meta II of G188C and the mutational analysis of the cysteine residue of channelrhodopsins, we revised the sentences in the “Results and discussion” section as follows.

What the reader also needs to know are the differences between dark state and Meta II which could be visualized by an overlay of the two available structures.

According to your comment, we constructed the structural models of the dark state and meta II of G188C mutant based on the 3D structures of the dark state and meta II of wild-type and superimposed these models to visualize the structural differences around position 188. We showed these structural models in Figure 1—figure supplement 1B.

Reviewer #2 (Recommendations for the authors):I support the publication without further modifications.

Thank you very much for your critical reading of our manuscript and for your valuable comments. I am very happy to know that you support the publication of the manuscript without further modifications.

Reviewer #3 (Recommendations for the authors):The data are thorough and convincing. However, the quality of presentation must be significantly improved.1. Overall the manuscript is too long, with numerous unnecessary repetitions. It can be shortened by 30-40% without losing the message.

According to your comment, we omitted a number of sentences, leaving the necessary explanations to help the understanding of non-specialist readers.

2. Evolutionary analysis (Figure S1) suggests that Ala, Ser, or Thr are in 188 position of most bistable opsins. How do the authors interpret this in the context of their data? The authors should discuss the difference in the properties of vertebrate opsin with Ala, Ser, or Thr are in 188 position (Figure S4), as compared to 188Cys? These data suggest that the residue in position 188 is not the only determinant of rhodopsin properties. The authors might want to revise Figure S7 accordingly.

Thank you very much for the important comment. Comparison of the amino acid sequences reveals that most bistable opsins have a threonine or serine residue at position 188 (Figure 1—figure supplement 1A). Thus, we speculate that an ancestral bistable opsin possessed a threonine or serine residue at position 188, which was mutated into the glycine residue in the common ancestor of mono-stable opsins. And the mutation at position 188 may have contributed to the loss of the bistable property and the acquisition of the mono-stable property of vertebrate rhodopsin. This is consistent with our recent finding that mutations at Thr188 of a bistable opsin, Opn5m, drastically hamper the bistable photoreaction (Fujiyabu et al., (2022) Commun. Biol. 5, 63). However, we could not create a bistable opsin by G188T or G188S mutation of bovine rhodopsin (Figure 2—figure supplement 1), which means that additional amino acid residues are necessary to explain the difference between the bistable and mono-stable property. To clarify this point, we added the following sentences in the “Results and discussion” section.

Page 14: “Comparison of the amino acid sequences reveals that most bistable opsins have a threonine or serine residue at position 188 (Figure 1—figure supplement 1A). Thus, we speculate that an ancestral bistable opsin possessed a threonine or serine residue at position 188, which was mutated into the glycine residue in the common ancestor of mono-stable opsins, including vertebrate rhodopsin, cone visual pigments and pinopsin. Also, the mutation at position 188 may have contributed to the change from the bistable property to the mono-stable property in the evolutionary ancestor of vertebrate rhodopsin. This is consistent with our recent finding that mutations at Thr188 of a bistable opsin, Opn5m, drastically hamper the bistable photoreaction. However, we could not create a bistable opsin by a single mutation, G188T or G188S, of bovine rhodopsin (Figure 2—figure supplement 1 and Table 1), which means that additional amino acid residue(s) are necessary to explain the difference between the bistable and mono-stable property from the viewpoint of the molecular evolution of opsins.”

3. Extensive editing, preferably by a native speaker, is needed. Some examples: line 35, "photoreceptions" should be "photoreception"; line 36, "sequences" should be "sequence"; line 50, "reversible with each other by light irradiations" should be "interconvertible by light"; lines 73, 86, 128, 148, 209, 223, delete "or not"; lines 73-74, "change the molecular property of bovine rhodopsin to acquire the photocyclic property" should be "can make bovine rhodopsin photocyclic"; line 91, "condition" should be "conditions"; line 125, "Altogether" should be "Collectively"; lines 152, 195 "irradiations" should be "irradiation"; lines 174-175, "overlapped that" should be "overlapped with that"; etc.

I thank you very much for your careful reading of the manuscript. We revised the manuscript according to all of your comments. Also, we tried to improve the English of our manuscript according to the advice from a native speaker of English.